# Improved Performance in the Control of DC-DC Three-Phase Power Electronic Converter Using Fractional-Order SMC and Synergetic Controllers and RL-TD3 Agent

**Marcel Nicola *** and **Claudiu-Ionel Nicola ***

Research and Development Department, National Institute for Research, Development and Testing in Electrical Engineering—ICMET Craiova, 200746 Craiova, Romania
* Correspondence: marcel_nicola@icmet.ro (M.N.); nicolaclaudiu@icmet.ro (C.-I.N.)

**Abstract:** In this article, starting from a benchmark represented by a Direct Current-to-Direct Current (DC-DC) three-phase power electronic converter used as an interface and interconnection between the grid and a DC microgrid, we compare the performances of a series of control structures—starting with the classical proportional integrator (PI) type and continuing with more advanced ones, such as sliding mode control (SMC), integer-order synergetic, and fractional-order (FO) controllers—in terms of maintaining the constant DC voltage of the DC microgrid. We present the topology and the mathematical modeling using differential equations and transfer functions of the DC-DC three-phase power electronic converter that provides the interface between the grid and a DC microgrid. The main task of the presented control systems is to maintain the DC voltage supplied to the microgrid at an imposed constant value, regardless of the total value of the current absorbed by the consumers connected to the DC microgrid. We present the elements of fractional calculus that were used to synthesize a first set of FO PI, FO tilt-integral-derivative (TID), and FO lead-lag controllers with Matlab R2021b and the Fractional-order Modeling and Control (FOMCON) toolbox, and these controllers significantly improved the control system performance of the DC-DC three-phase power electronic converter compared to classical PI controllers. The next set of proposed and synthesized controllers were based on SMC, together with its more general and flexible synergetic control variant, and both integer-order and FO controllers were developed. The proposed control structures are cascade control structures combining the SMC properties of robustness and control over nonlinear systems for the outer voltage control loop with the use of properly tuned synergetic controllers to obtain faster response time for the inner current control loop. To achieve superior performance, this type of cascade control also used a properly trained reinforcement learning-twin delayed deep deterministic policy gradient (RL-TD3) agent, which provides correction signals overlapping with the command signals of the current and voltage controllers. We present the Matlab/Simulink R2021b implementations of the synthesized controllers and the RL-TD3 agent, along with the results of numerical simulations performed for the comparison of the performance of the control structures.

**Keywords:** power electronic converter; fractional order; sliding mode control; synergetic control; reinforcement learning

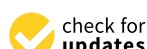

## 1. Introduction

Due to the expansion and improvement of technologies for the construction of renewable energy power generation systems, there are growing numbers of both AC and DC microgrids. Although each of these microgrids has its own particularities, the types of topologies used can represent a significant number of the variants in the construction of microgrids, and we can identify some elements of great importance in the microgrid architecture, one of them being, of course, the DC-AC or DC-DC converter [1–10].

A remarkable type of DC-DC converter is the multi-phase interleaved converter, and one of its basic characteristics is that the failure of components in one phase does not affect

the operation of the rest of the converter phases. Thus, this type of DC-DC converter is highly reliable and can be successfully used as an interconnection interface between a main grid (which also contains an AC-DC converter) and a DC microgrid [11–15].

As far as the control system of such a DC-DC power electronic converter is concerned, we can say that it is usually built using classic PI controllers [16–19]. The controllers synthesized using SMC theory have a special role in this class of controllers, since they are robust and suitable for the control of nonlinear systems [20,21]. A generalization of SMC controllers—namely, synergetic controllers [22,23]—has also been developed. They retain the robustness of SMC controllers but, with an additional degree of freedom, they can provide superior response time performance. Thus, for a cascade control system, it is necessary to use an SMC controller for outer loop control and a synergetic controller for inner loop control.

In this article, we start from a benchmark used in [18,19] consisting of a DC-DC three-phase power electronic converter, which allows the comparison of the performance of the control systems of this type of converter when their main task is to maintain the constant voltage supplied to a DC microgrid under the conditions of variable consumption required by it. We propose and synthesize a series of controllers with which superior performance can be obtained for the benchmark presented.

In order to improve the performance of the control systems used with the mentioned benchmark, in this article, starting from the definitions and particular structures of fractional calculus, we propose and synthesize a series of fractional and integer controllers, as well as combined controllers to be used in voltage outer loop control and current inner loop control [24,25]. A properly trained RL-TD3 agent combined with a cascade control structure is also used to improve this performance [26,27]. The main performance indicators covered in the comparison of the power electronic converter control systems are: steady-state error, overshoot, response time, and the ripple in the DC voltage supplied to the DC microgrid.

Among the main contributions of this article, we can mention:

- Mathematical modeling using differential equations and transfer functions of the power electronic converter that provides the interface and interconnection between the grid and a DC microgrid, a system that is used as a benchmark;
- Synthesis of fractional controllers using Matlab R2021b and the FOMCON toolbox [28–30]—i.e., FO PI, FO TID, and FO lead-lag—for the presented benchmark;
- Synthesis of both integer and fractional SMC and synergetic controllers for the presented benchmark;
- Implementation in Matlab/Simulink R2021b of a control structure proposed by the authors consisting of FO SMC controllers for voltage outer loop control and FO synergetic controllers for current inner loop control operating in tandem with an RL-TD3 agent to achieve superior control performance with the presented benchmark;
- Implementation in Matlab/Simulink of the synthesized controllers in order to compare the performance of the control system with the presented benchmark.

The rest of the paper is structured as follows: the mathematical modeling of the power electronic converter between two DC microgrids is presented in Section 2. The FO PI, FO TID, and FO lead-lag controllers for the voltage outer loop control are presented in Section 3, while the SMC and synergetic controller—both integer-order and FO—combined with the RL-TD3 agent are presented in Section 4. In Section 5, the implementation of these control structures using the Matlab/Simulink R2021b environment and the comparison of their performances through numerical simulations are described for the benchmark presented. Conclusions and clarifications about future approaches are presented in the final section.

## 2. Mathematical Modeling of the Power Electronic Converter between Two DC Microgrids

Starting from the topology of the two-way power electronic converter described in [18,19] and shown in Figure 1, this section presents the equations and transfer functions needed to model the operation of the three-phase converter in order to obtain a

common benchmark to compare the control methods and types proposed in [18,19] with the controllers proposed in this paper, which are based on fractional calculus, as well as improvements using the RL-TD3 agent; these controllers are described gradually in Sections 3 and 4. Therefore, the main task of the control system for the power electronic converter with this benchmark is to maintain the constant $V_{DC}$ voltage for the supply of the DC microgrid (on DC bus 2) under conditions where the current absorbed by the consumers connected to this microgrid, represented by the current $i_0$, can have significant variations.

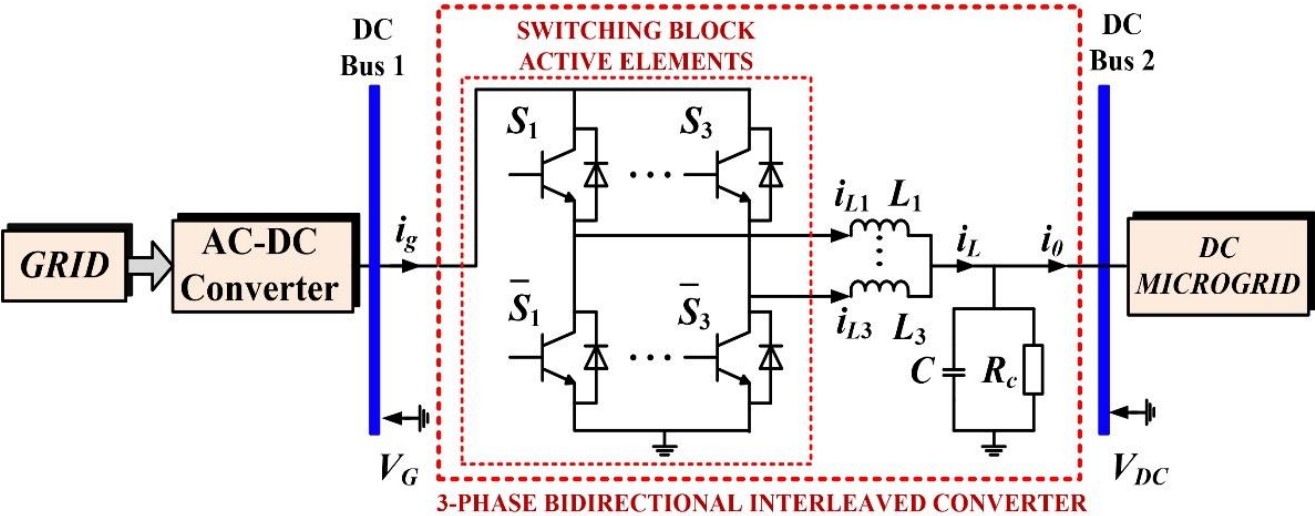

**Figure 1.** Block diagram of the three-phase power electronic converter.

In Figure 1, the DC voltage on DC bus 1 is denoted $V_G$, and the modulation indices of the control signals $S_k$ ($k$ = 1, 2, 3) of the active switching elements are denoted $m_k$ ($k$ = 1, 2, 3). Accordingly, Figure 2 shows an equivalent circuit for the power electronic converter. We used the typical notations for the modelling of the inductances, resistances, and capacitances in the diagram shown.

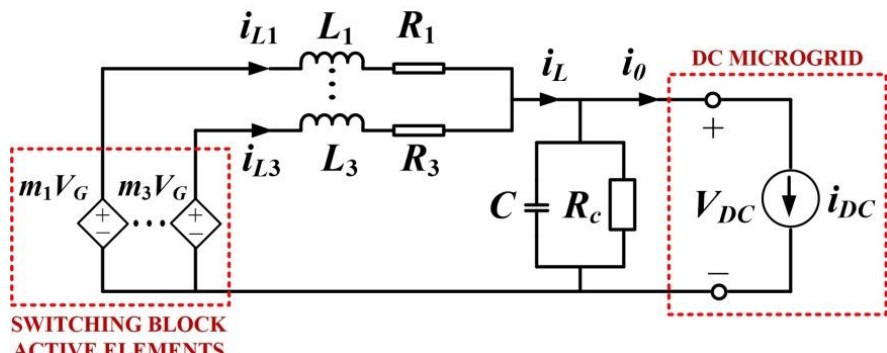

**Figure 2.** Equivalent block diagram of the three-phase power electronic converter.

By applying Kirchhoff's laws to the system in Figure 2, the following equations can be obtained:

$$- m_k V_G + L_k \frac{di_{Lk}}{dt} + R_k i_{Lk} + V_{DC} = 0 \qquad (1)$$

$$\sum_{k=1}^{3} i_{Lk} - i_0 - C \frac{dV_{DC}}{dt} = 0 \qquad (2)$$

If the above equations are written in the time domain to achieve the modeling using transfer functions, the Laplace transform can be applied, and the following equations are obtained:

$$-m_k(s)V_G + (L_k + R_k)I_{Lk}(s) + V_{DC}(s) = 0 \tag{3}$$

$$\sum_{k=1}^{3} I_{Lk}(s) - I_0(s) - CsV_{DC}(s) = 0 \tag{4}$$

Without compromising the generality and simply for ease of writing and schematic representation, we consider that the circuit elements in each phase are equal; i.e., $L_1 = L_2 = L_3 = L$ and $R_1 = R_2 = R_3 = R$. Accordingly, the following equation is obtained by summation from the last equations:

$$-\sum_{k=1}^{3} m_k(s)V_G + (Ls + R)\sum_{k=1}^{3} I_{Lk}(s) + 3V_{DC}(s) = 0 \tag{5}$$

First, we assume that the value of current $i_0$ is equal to 0. From the point of view of the modeling using transfer functions, this current can be represented as a disturbance for the control systems of the microgrid $V_{DC}$ voltage. For phase $k$ and for non-zero $m_k$, the following relation can be written:

$$-m_k(s)V_G + (Ls + R)\sum_{k=1}^{3} I_{Lk}(s) + 3V_{DC}(s) = 0 \tag{6}$$

By rewriting Equation (4), we obtain:

$$\sum_{k=1}^{3} I_{Lk}(s) = CsV_{DC}(s) \tag{7}$$

By substituting Equation (6) into Equation (7), we obtain the following equation:

$$-m_k(s)V_G + \left(CLs^2 + CRs + 3\right)V_{DC}(s) = 0 \tag{8}$$

By using Equation (3), the following relation can be written:

$$V_{DC}(s) = m_k(s)V_G - (Ls + R)I_{Lk}(s) \tag{9}$$

As a result, we obtain the transfer function shown in Figure 3 in the following form:

$$\frac{I_{Lk}(s)}{m_k(s)} = \frac{V_G}{(Ls + R)} \frac{(CLs^2 + CRs + 2)}{(CLs^2 + CRs + 3)} \tag{10}$$

**Figure 3.** Block diagram modeled using the transfer function of the control system for the power electronic converter.

By using Equation (4), we obtain the following relation:

$$3I_{Lk}(s) - I_0(s) - CsV_{DC}(s) = 0 \tag{11}$$

and, hence, the transfer function from the last block in Figure 3 as:

$$\frac{V_{DC}(s)}{I_{Lk}(s)} = \frac{3}{Cs} \tag{12}$$

It can be seen from Figure 3 that, to equate the control system of the power electronic converter for each phase by means of transfer functions, we use a cascade control system with an inner regulating loop of $i_{Lref}$ current prescribed by the controller of the outer regulating loop of $V_{DC}$ voltage.

The notations are the usual ones for the representation of the transformation of current and voltage quantities by means of sensors—the transformation factors $I_{base}$ and $V_{base}$—and PI-type current and voltage controllers with proportionality factors $K_{pv}$ and $K_{pc}$. The integration factors are denoted $K_{iv}$ and $K_{iv}$ for the voltage and current control loops, respectively.

Simplifying the last term in Equation (10) by equating it with the unit from [18,19], we obtain the equivalent diagram using the transfer functions of the inner current control loop shown in Figure 4.

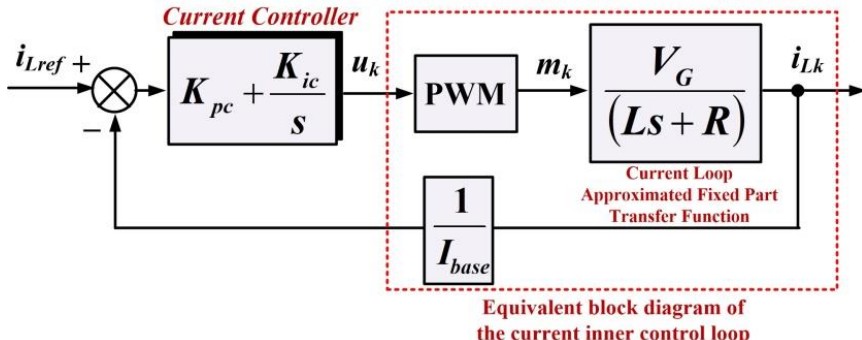

**Figure 4.** Equivalent block diagram modeled using the transfer function of the inner loop for current regulation.

To compare the various control structures using the benchmark from [18,19], for the two types of PI controllers mentioned, we use as prompts the methods proposed in [18,19]; namely, tuning with the Gao method described in [17] and its improvement described in [18]. Thus, considering the bandwidth $\omega_c$ (for the inner current control loop), the authors of [18] proposed tuning relations as follows:

$$K_{pc} = \frac{\omega_c L I_{base}}{V_G} \tag{13}$$

$$K_{ic} = \frac{\omega_c R I_{base}}{V_G} \tag{14}$$

Accordingly, the approximation of the current control system (the inner loop) is obtained as follows:

$$\frac{I_{Lk}}{I_{Lref}} = \frac{\omega_c}{s + \omega_c} \tag{15}$$

By tuning the PI-type voltage controller using Gao's method described in [18,19], we can, under the condition that a bandwidth $\omega_v$ (for the outer voltage control loop) is required, use the following relations:

$$K_{pv} = \frac{\omega_v C V_{base}}{3 I_{base}} \tag{16}$$

$$K_{iv} = \frac{\omega_v V_{base}}{3 R_c I_{base}} \tag{17}$$

Further, as we have drawn on [18] for the tuning of the PI-type current and voltage controllers proposed using Equations (14)–(17), we name them PI-Gao controllers for short. For the improvement of the performance, we tune the controllers according to a relation in the form shown in Equation (18), and we name them PI-Gamma controllers for short ($\gamma$):

$$K_{iv\gamma} = \frac{\gamma \omega_v C V_{base}}{3 I_{base}} \tag{18}$$

where $\gamma$ is a tuning parameter that depends on $\omega_c$.

With these, it is possible to model the influence of the current $i_0$ on the voltage $V_{DC}$ using the following transfer function:

$$\frac{V_{DC}(s)}{I_0(s)} = \frac{\frac{1}{Cs}}{1 + \frac{3 I_{base}}{V_{base}} \frac{Y(s)}{Cs}} \tag{19}$$

where we can note:

$$Y(s) = \left( K_{pv} + \frac{K_{iv\gamma}}{s} \right) \frac{\omega_c}{s + \omega_c} \tag{20}$$

In the following section, based on the general block diagram shown in Figure 5, we develop FO-type controllers—i.e., FO PI, FO TID, and FO lead-lag controllers—for the performance comparison. It can be seen in Figure 5 that, for these controllers, the inner current control structure is approximated as described above with a transfer function of the first order.

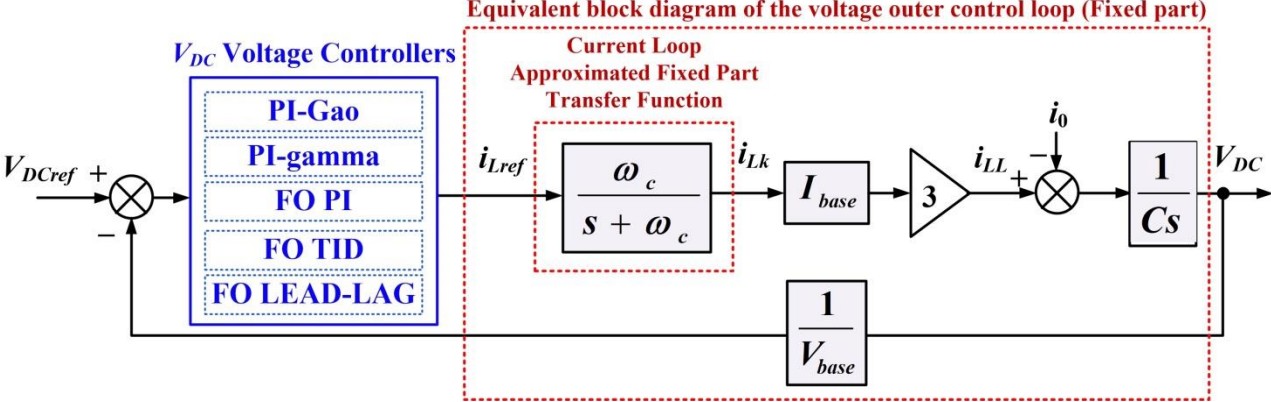

**Figure 5.** Block diagram modeled using the transfer function of the control system for the power electronic converter with PI-Gao, PI-gamma, FO PI, FO TID, and FO lead-lag voltage controllers.

The basic parameters of the power electronic converter based on which the comparisons and numerical simulations of the control systems were performed are presented in Table 1. It can be noted that, in accordance with the choice of the bandwidths of the two control loops, the response time of the inner current control loop is ten times shorter than the response time of the inner $V_{DC}$ voltage control loop.

**Table 1.** Nominal parameters for the power electronic converter.

| Parameter | Value | Unit |
|---|---|---|
| Input voltage from the DC grid $V_G$ (DC bus 1) | 360 | V |
| Output DC microgrid voltage $V_{DC}$ (DC bus 2) | 400 | V |
| Inductance $L$ | 2.5 | mH |
| Capacitor $C$ | 1.175 | mF |
| Resistance $R_c$ | 0 | $\Omega$ |
| Voltage sensor constant transform parameters $V_{base}$ | 200 | V |
| Current sensor constant transform parameters $I_{base}$ | 28 | A |
| Bandwidth of the outer voltage loop regulation $\omega_v$ | $100\cdot\pi$ | rad/s |
| Bandwidth of the inner current loop regulation $\omega_c$ | $1000\cdot\pi$ | rad/s |

We should specify that, for the parameters of the PI-Gao and PI-gamma controllers in the comparisons of the performance of the control systems, we use the parameters obtained from the above relations and presented in detail in [18,19]; i.e., for PI-Gao, we obtain $K_{pv} = 0.8789$ and $K_{iv} = 0.0159$; for PI-gamma with $\gamma = \omega_c/100$, we obtain $K_{pv} = 0.8789$ and $K_{iv} = 27.6114$.

## 3. Fractional-Order Controllers for Voltage Outer Loop Control

In accordance with [28–30], we here present elements of differential and integral fractional-order calculus in which a central role is played by operator $aD_t^\alpha$, as described by the following relation:

$$aD_t^\alpha = \begin{cases} \frac{d^\alpha}{dt^\alpha} & \text{Re}(\alpha) > 0 \\ 1 & \text{Re}(\alpha) = 0 \\ \int_a^t (dt)^{-\alpha} & \text{Re}(\alpha) < 0 \end{cases} \tag{21}$$

We should specify that the notations used in Equation (21) have the following significance: $a$ and $t$ represent the limits of the range to which the operator is applied, while the term $\alpha$ represents the fractional order. The Riemann–Liouville definition, which should be useful in the numerical implementation of the results obtained, is presented in Equation (22):

$$aD_t^\alpha f(t) = \frac{1}{\Gamma(m - \alpha)} \left(\frac{d}{dt}\right)^m \int_\alpha^t \frac{f(\tau)}{(t - \tau)^{\alpha - m + 1}} d\tau \tag{22}$$

where $m - 1 < \alpha < m$; $m \in N$; and $\Gamma(\cdot)$ is the notation for the Euler gamma function.

### 3.1. FO PI-Type Controller

The authors of [28–30] present a gradual introduction to the Laplace transform and fractional-order transfer functions, which we draw from in Equation (23) to show the general form of a fractional PID controller, denoted $PI^\lambda D^\mu$, while $u(t)$ and $e(t)$ are the output and input of the controller:

$$u(t) = K_p e(t) + K_i D^{-\lambda} e(t) + K_d D^\mu e(t) \tag{23}$$

Accordingly, the general form expressed by a transfer function of the fractional-order PI controller denoted FO PI is given in Equation (24):

$$H_{FO-PI}(s) = K_p + \frac{K_i}{s^\lambda} \tag{24}$$

where $K_p$ denotes the proportional factor, $K_i$ denotes the integral factor, $\lambda$ (positive) denotes the order of the integrator, $K_d$ denotes the differential coefficient, and $\mu$ denotes the order of the differentiator. It can be noted that, when $\lambda = \mu = 1$, we obtain the usual integer-order PID controller.

To work with fractional-order transfer functions, we used the FOMCON toolbox for Matlab R2021b [29,30]. The integer transfer function for the fixed part of the power electronic converter control system is given in Equation (25):

$$H_{fixed\_part}(s) = \frac{2.639 \cdot 10^5}{0.001175 \cdot s^2 + 3.691 \cdot s} \tag{25}$$

The transfer function given in Equation (24), following an integral absolute error (IAE) criterion optimization process performed with the FOMCON toolbox, takes the form given in Equation (26), where the FO PI controller parameters have the following values: $K_p = 3$, $K_i = 1.8$, and $\lambda = 4.48$.

$$H_{F0-PI}(s) = \frac{3 \cdot s^{4.48} + 1.8}{s^{4.48}} \tag{26}$$

In the closed loop, when using an FO PI controller, the transfer function takes the form given in Equation (27):

$$H_{CL\_FO-PI} = \frac{3 \cdot s^{4.48} + 1.8}{0.001175 \cdot s^{6.49} + 3.691 \cdot s^{5.49} + 3 \cdot s^{4.48} + 1.8} \tag{27}$$

Obviously, in order to implement the transfer function of the fractional-order controller in an embedded system, an integer approximation of this transfer function must be obtained. Thus, in accordance with [28–30], we can define the approximation of a fractional transfer function for a frequency range $(\omega_b, \omega_h)$, an order $N$, and $s^\beta$ ($0 < \beta < 1$) by using Oustaloup recursive filters in the form of Equations (28) and (29):

$$G_f(s) = K \prod_{k=-N}^{N} \frac{s + \omega_k'}{s + \omega_k} \tag{28}$$

where, for the values of $\omega_k'$, $\omega_k$, and $K$, the following relations can be written:

$$\omega_k' = \omega_b \left( \frac{\omega_h}{\omega_b} \right)^{\frac{k+N+\frac{1}{2}(1-\beta)}{2N+1}} ; \ \omega_k = \omega_b \left( \frac{\omega_h}{\omega_b} \right)^{\frac{k+N+\frac{1}{2}(1+\beta)}{2N+1}} ; \ k = \omega_h^\beta \tag{29}$$

A refined version of the Oustaloup recursive filters is given by the following relations [28–30]:

$$s^\alpha \approx \left( \frac{d\omega_h}{b} \right)^\alpha \left( \frac{ds^2 + b\omega_h s}{d(1-\alpha)s^2 + b\omega_h s + d\alpha} \right) G_p \tag{30}$$

$$G_p = K \prod_{k=-N}^{N} \frac{s + \omega_k'}{s + \omega_k} ; \ \omega_k = \left( \frac{b\omega_h}{d} \right)^{\frac{\alpha+2k}{2N+1}} ; \ \omega_k' = \left( \frac{d\omega_b}{b} \right)^{\frac{\alpha-2k}{2N+1}} \tag{31}$$

Note that, in Equations (30) and (31), the usual values are: $b = 10$ and $d = 9$.

### 3.2. FO TID Controller

Another type of fractional-order controller used in process-control applications is the TID controller, the general form of which, expressed by the transfer function, is given in Equation (32):

$$H_{FO-TID}(s) = \frac{K_t}{s^{1/n}} + \frac{K_i}{s} + K_d s \tag{32}$$

where $K_t$ denotes the tilt gain, $n$ denotes the order of integration of the term tilt, $K_i$ denotes the amplification factor of the integrator term, and $K_d$ denotes the amplification factor of the derivation term.

The transfer function of the FO TID controller has the following form:

$$H_{FO-TID}(s) = \frac{1.2 \cdot s^{0.91} + 12}{s} \tag{33}$$

When using the FOMCON toolbox with a closed loop, the transfer function in the case of an FO TID controller takes the form given by Equation (34):

$$H_{CL-FO-TID}(s) = \frac{316,800 \cdot s^{0.91} + 3,166,800}{0.001175 \cdot s^{3.01} + 3.691 \cdot s^{2.01} + 316,800 \cdot s^{0.91} + 3,166,800} \tag{34}$$

### 3.3. FO Lead-Lag Controller

The next fractional-order controller proposed to improve the performance of the power electronic converter control system is the FO lead-lag controller. The general form of the transfer function for an FO lead-lag controller is given by Equation (35):

$$H_{FO-Lead-Lag}(s) = K_c \left( \frac{s + \frac{1}{\lambda}}{s + \frac{1}{x\lambda}} \right)^{\alpha} = K_c x^{\alpha} \left( \frac{\lambda s + 1}{x\lambda s + 1} \right)^{\alpha}, \ 0 < x < 1 \tag{35}$$

where $\lambda$ denotes the fractional order of the FO lead-lag controller.

From the form of the transfer function for the FO lead-lag controller, it can be noted that a lead effect is obtained when $\alpha > 0$, while a lag effect is obtained when $\alpha < 0$.

With $k' = K_c x^{\alpha}$, we obtain the usual form of this controller:

$$H_{FO-Lead-Lag}(s) = k' \left( \frac{\lambda s + 1}{x\lambda s + 1} \right)^{\alpha} \tag{36}$$

It can be noted that, for $k' = \alpha = 1$, $\lambda = \frac{K_p}{K_i}$, and a very large value of $x$ (for example, $x > 10,000$), the transfer function of the FO lead-lag controller becomes the transfer function of the FO PI controller. It can, therefore, be concluded that the use of the FO lead-lag controller in a control loop has great flexibility.

The transfer function of the FO lead-lag controller is given in the following form:

$$H_{FO-Lead-Lag}(s) = \frac{1.8023 \cdot s^{2.2} + 1.4201 \cdot s^{1.1} + 7.024}{s^{2.2} + 2.196 \cdot s^{1.1} + 1} \tag{37}$$

In a closed loop, the transfer function in the case of an FO lead-lag controller takes the form given by Equation (38):

$$H_{CL\_FO-Lead-Leag}(s) =$$

$$\frac{475,630 \cdot s^{2.2} + 374,760 \cdot s^{1.1} + 1,853,600}{0.00117 \cdot s^{4.21} + 3.691 \cdot s^{3.21} + 0.002583 \cdot s^{3.11} + 47,563 \cdot s^{2.2} + 8.1054 \cdot s^{2.11} + 0.00117 \cdot s^{2.01} + 374,760 \cdot s^{1.1} + 3.691 \cdot s^{1.01} + 1,853,600} \tag{38}$$

Figure 6 shows the stability test for the closed-loop control system for the power electronic converter based on the FO lead-lag controller for the control of voltage $V_{DC}$. It can be noted that the system is stable. Figure 7 shows the Bode diagram of the closed-loop control system. The system is stable because the following condition is fulfilled: $\omega_t < \omega_{\pi}$, where the notations are the usual ones; i.e., $\omega_t$ represents the zero-crossing frequency of the logarithmic amplitude frequency characteristic, while $\omega_{\pi}$ represents the frequency for which the phase response is equal to $-\pi$ rad/s.

In Appendix A is presented the modality to obtain the integer transfer functions for the FO PI, FO TID, and FO lead-lag controllers in continuous and discrete form.

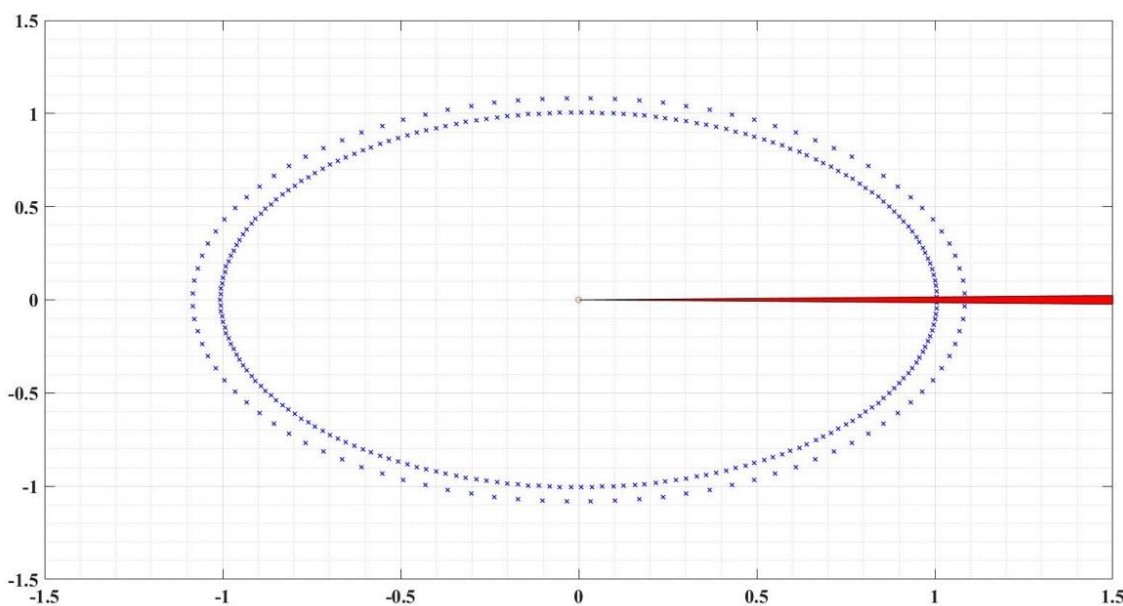

**Figure 6.** Stability test for the closed-loop control system for the power electronic converter using an FO lead-lag controller.

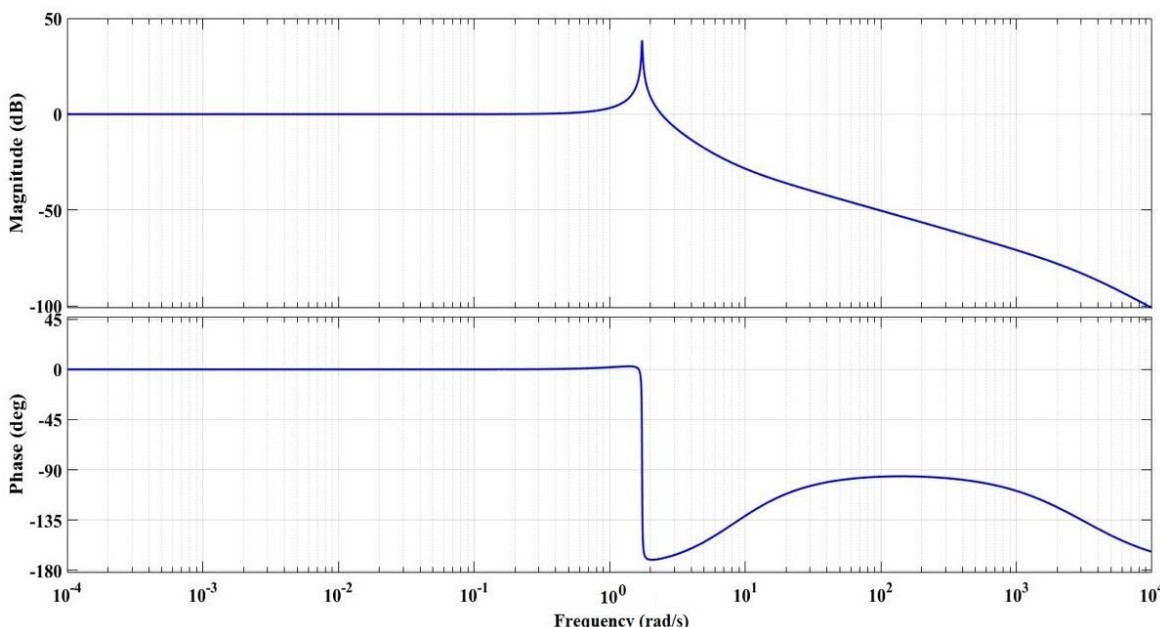

**Figure 7.** Bode diagram for the closed-loop control system for the power electronic converter using an FO lead-lag controller.

## 4. Fractional-Order SMC and Synergetic Controllers Combined with RL-TD3 Agent

Starting from Equation (11), where $i_0$ represents the current absorbed by the consumers connected to the DC microgrid (on DC bus 2), it can be seen that, in order to implement this in the $V_{DC}$ voltage control system (the outer loop in Figure 3), there must be an inner loop for the control of the current absorbed by the consumers. This can be achieved by adding the variable $i_{Lref}$ supplied by the voltage controller through the imposition of the condition $i_0 = i_{Lref}$, based on which we can write the following equation:

$$C\frac{dV_{DC}}{dt} = \sum_{k=1}^{3} i_{Lk} - i_{Lref} \tag{39}$$

Next, we synthesize an SMC controller and then an FO SMC controller for which the $i_{Lref}$ output is the reference for the inner current control loop. In Section 4.1, we describe the approximation employed in Sections 2 and 3 in terms of the approximation of the inner current control loop by means of a transfer function of the first order.

### 4.1. Fractional-Order SMC Controller for Voltage Outer Loop Control

For the synthesis of the SMC controller, we define the state variable $x_1$ as follows:

$$x_1 = V_{DCref} - V_{DC} \tag{40}$$

We also define the switching surface $S$:

$$\begin{cases} S = c_1 x_1 + x_2 \\ \dot{S} = c_1 x_2 + \dot{x}_2 \end{cases} \tag{41}$$

where we define the state variable $x_2$ as follows:

$$x_2 = \dot{x}_1 = -\dot{V}_{DC} \tag{42}$$

By choosing a Lyapunov function $V = \frac{S^2}{2}$, according to the condition that $\dot{V} = S \cdot \dot{S} < 0$, it is obvious that, to ensure convergence, the following relation is required:

$$\dot{S} = -\varepsilon \operatorname{sgn}(S) - kS \tag{43}$$

where $\varepsilon$ and $k$ are positive constants.

From the calculations, we obtain:

$$\ddot{x}_1 = \dot{x}_2 = -\ddot{V}_{DC} = \frac{1}{C}\dot{i}_{Lref} - \frac{1}{C}\sum_{k=1}^{3} \dot{i}_{Lk} \tag{44}$$

From Equation (41), (43), and (44), we obtain $i_{Lref}$, the SMC controller output, in the following form:

$$i_{Lref} = C\int_0^t \left[ -(c_1 x_2 + kS + \varepsilon h(S)) + \frac{1}{C}\sum_{k=1}^{3} \dot{i}_{Lk} \right] dt \tag{45}$$

Figure 8 presents a concise block diagram modeled using the transfer function for the control system of the power electronic converter using an SMC controller for the voltage outer control loop. For the inner current control loop, we use the approximation with a transfer function of the first order.

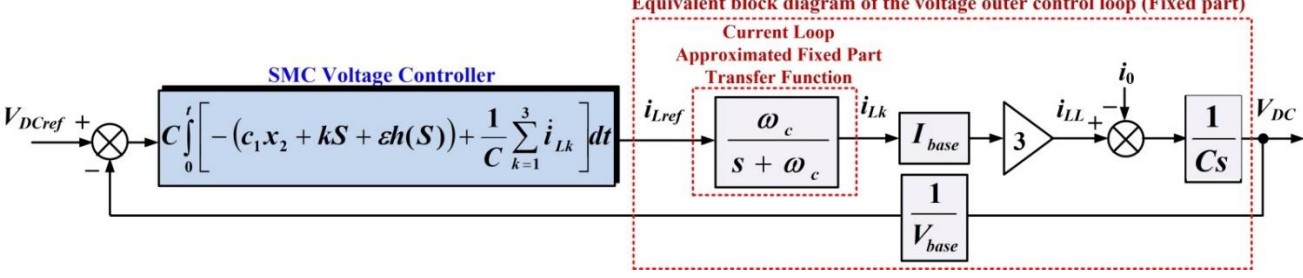

**Figure 8.** Block diagram modeled using the transfer function for the control system of the power electronic converter using an SMC controller for the voltage outer control loop.

For the fractional case, for the synthesis of an FO SMC controller, the switching surface is defined as follows:

$$S = c_1 x_1 + c_2 D^{\mu} x_1 = c_1 x_1 + c_2 D^{\mu-1} x_2 \tag{46}$$

where the fractional differential operator $D$ is defined by Equation (21).

By calculating the derivative of this surface $\dot{S}$, we obtain the following relation:

$$\dot{S} = c_1 \dot{x}_1 + c_2 D^{\mu+1} x_1 = c_1 x_2 + c_2 D^{\mu-1} \dot{x}_2, \tag{47}$$

which can be rewritten using Equation (44) as follows:

$$\dot{S} = c_1 x_2 + c_2 D^{\mu-1} \left( \frac{1}{C} \dot{i}_{Lref} - \frac{1}{C} \sum_{k=1}^{3} \dot{i}_{Lk} \right) \tag{48}$$

Using a Lyapunov function, as in the case of the integer-order SMC controller, and Equation (43), we obtain the following relation:

$$- \varepsilon h(S) - kS - c_1 x_2 = c_2 D^{\mu-1} \left( \frac{1}{C} \dot{i}_{Lref} - \frac{1}{C} \sum_{k=1}^{3} \dot{i}_{Lk} \right) \tag{49}$$

By applying the operator $D^{1-\mu}$, according to the definition, to the previous equation, we obtain the following relation:

$$D^{1-\mu} \left( - \varepsilon h(S) - kS - c_1 x_2 \right) = c_2 \left( \frac{1}{C} \dot{i}_{Lref} - \frac{1}{C} \sum_{k=1}^{3} \dot{i}_{Lk} \right) \tag{50}$$

From this, we derive the output $i_{Lref}$ of the FO SMC controller as follows:

$$i_{Lref} = \frac{C}{c_2} \int_0^t \left[ c_2 \frac{1}{C} \sum_{k=1}^{3} \dot{i}_{Lk} + D^{1-\mu} \left( - \varepsilon h(S) - kS - c_1 x_2 \right) \right] dt \tag{51}$$

Figure 9 presents the concise block diagram modeled using the transfer function for the control system of the power electronic converter using an FO SMC controller for the voltage outer control loop. For the inner current control loop, we use the approximation with a transfer function of the first order.

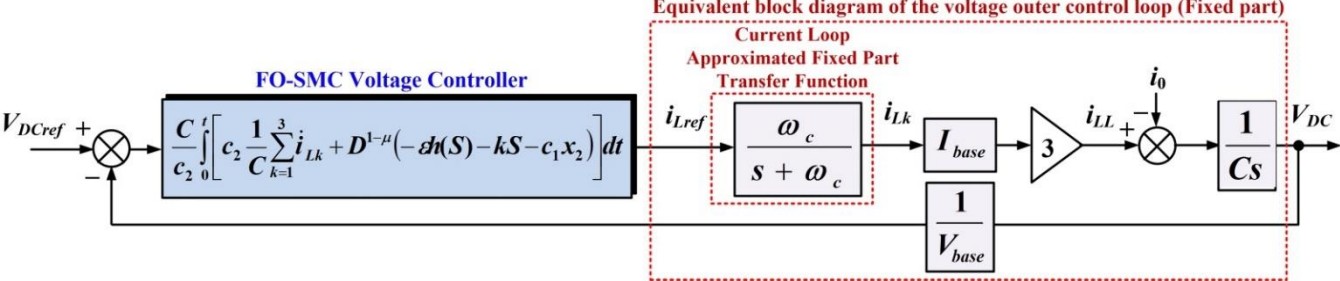

**Figure 9.** Block diagram modeled using a transfer function for the control system of the power electronic converter using an FO SMC controller for the voltage outer control loop.

Due to the fact that, in Figures 8 and 9, the approximation of the inner current control loop with a transfer function of the first order is achieved by using PI-type controllers for the current control loops and the approximation relations of the type shown in Equations (13) and (14), in order to compare the performance of these control structures with those from Figures 6 and 7, they will be named the SMC controller and PI controller and the FO SMC controller and PI controller, respectively, for short.

### 4.2. Fractional-Order Synergetic Controller for Current Inner Loop Control

Since, in the case of the cascade control, the inner control loop must be faster than the outer control loop, in this section we consider SMC and FO SMC controllers for the outer voltage control loop. For the inner current control loop, we synthesize synergetic and FO synergetic controllers. This is because these types of controllers are also part of the sliding control category but, due to the particular way they are described, they have an additional design parameter that gives them better flexibility and response time compared to SMC type controllers.

The general form of a synergetic controller can be written as:

$$\dot{x} = f(x, u, t) \tag{52}$$

where $x$ denotes the state vector and $x \in \Re^n$, $f(.)$ denotes the continuous nonlinear description function, and $u$ denotes the control vector and $u \in \Re^m$ with $m < n$.

In the synergetic controller synthesis algorithm, for each control input, we choose a macro-variable $\Psi(x,t)$, which is dependent on the states of the system.

The synthesis algorithm entails the evolution of the system to a differential manifold $\Psi = 0$ according to the following equation:

$$T\dot{\psi} + \psi = 0 \tag{53}$$

where $T > 0$ denotes a parameter that dictates the rate of convergence to the differential manifold $\Psi(x,t)$.

By deriving the macro-variable $\Psi$, we can write the following relation:

$$\dot{\psi} = \frac{\partial \psi}{\partial x} \dot{x} \tag{54}$$

By substituting Equation (54) into Equation (53), we obtain the following relation:

$$T\frac{\partial \psi}{\partial x}\dot{x} + \psi = 0 \tag{55}$$

In order to obtain the control law, we must include the explicit forms of the state variables $\dot{x}$ deduced from the mathematical model formulated for the controlled system, which can be written in the following general form:

$$u = u(x, \psi(x, t), T, t) \tag{56}$$

Next, we apply the synergetic integer- and fractional-order control procedure to replace the PI-type controller in the inner current control loop, and the outputs of each inner current control loop corresponding to each phase are applied to the $m_k$ modulation indices. The $S_k$ ($k = 1, 2, 3$) pulses are obtained from these by means of PWM blocks and act on the IGBT-type active element controls for the DC-DC converter shown in Figure 1.

Next, for $k^* > 0$, we choose a macro-variable $\Psi$ with the following form:

$$\psi = \left( V_{DCref} - V_{DC} \right) + k^* \left( i_{Lref} - \sum_{k=1}^{3} i_{Lk} \right) \tag{57}$$

In addition to the state variable $x_1$ defined in Equation (40) for the synthesis of the SMC controller, a state variable $x_2$ can be defined as in the following relation:

$$\begin{cases} x_1 = V_{DCref} - V_{DC} \\ x_2 = i_{Lref} - \sum\limits_{k=1}^{3} i_{Lk} \end{cases} \tag{58}$$

It can be noted that, in Equation (58), for a quasi-steady-state regime or for slow variations in the reference quantities, we can consider their derivative as null and, thus, Equation (58) becomes:

$$\begin{cases} \dot{x}_1 = -\dot{V}_{DC} \\ \dot{x}_2 = -\sum_{k=1}^{3} \dot{i}_{Lk} \end{cases} \tag{59}$$

By using Equation (59) for the calculation of the derivative of the macro-variable $\Psi$, we obtain the following relation:

$$\dot{\psi} = \dot{x}_1 + k^* \dot{x}_2 = -\dot{V}_{DC} - k^* \sum_{k=1}^{3} \dot{i}_{Lk} \tag{60}$$

By adapting Equation (55) for the elements of the presented system, we obtain:

$$T\left(-\dot{V}_{DC} - k^* \sum_{k=1}^{3} \dot{i}_{Lk}\right) + \left(V_{DCref} - V_{DC}\right) + k^* \left(i_{Lref} - \sum_{k=1}^{3} i_{Lk}\right) = 0 \tag{61}$$

Next, we can use Equation (1), in which, for ease of writing, we denote the control inputs as:

$$u_k = -m_k V_G, \ k = 1, 2, 3 \tag{62}$$

Accordingly, we obtain the following relation:

$$L_k \frac{di_{Lk}}{dt} = u_k - R_k i_{Lk} - V_{DC} \tag{63}$$

Furthermore, the following notations are used for ease of writing:

$$u_{3k} = -R_k i_{Lk} - V_{DC}, \ k = 1, 2, 3 \tag{64}$$

Accordingly, we obtain the relations in Equation (65):

$$L_k \frac{di_{Lk}}{dt} = u_k + u_{3k}, \ k = 1, 2, 3 \tag{65}$$

By substituting this equation into Equation (61), we can write:

$$-T\dot{V}_{DC} - Tk^* \frac{1}{L_k} (u_{3k} + u_k) + \left(V_{DCref} - V_{DC}\right) + k^* \left(i_{Lref} - \sum_{k=1}^{3} i_{Lk}\right) = 0 \tag{66}$$

After rearranging the terms in Equation (66), we obtain:

$$Tk^* \frac{1}{L_k} u_k = -T\dot{V}_{DC} - Tk^* \frac{1}{L_k} u_{3k} + \left(V_{DCref} - V_{DC}\right) + k^* \left(i_{Lref} - \sum_{k=1}^{3} i_{Lk}\right) \tag{67}$$

The control law $u_k$ for $k = 1, 2, 3$ can be obtained from Equation (67) as follows:

$$u_k = \frac{L_k}{Tk^*} \left[-T\dot{V}_{DC} - Tk^* \frac{1}{L_k} u_{3k} + \left(V_{DCref} - V_{DC}\right) + k^* \left(i_{Lref} - \sum_{k=1}^{3} i_{Lk}\right)\right] \tag{68}$$

Figure 10 presents the concise block diagram modeled using the transfer function for the control system of the power electronic converter using an SMC controller for the outer voltage control loop and a synergetic controller for the inner current control loop.

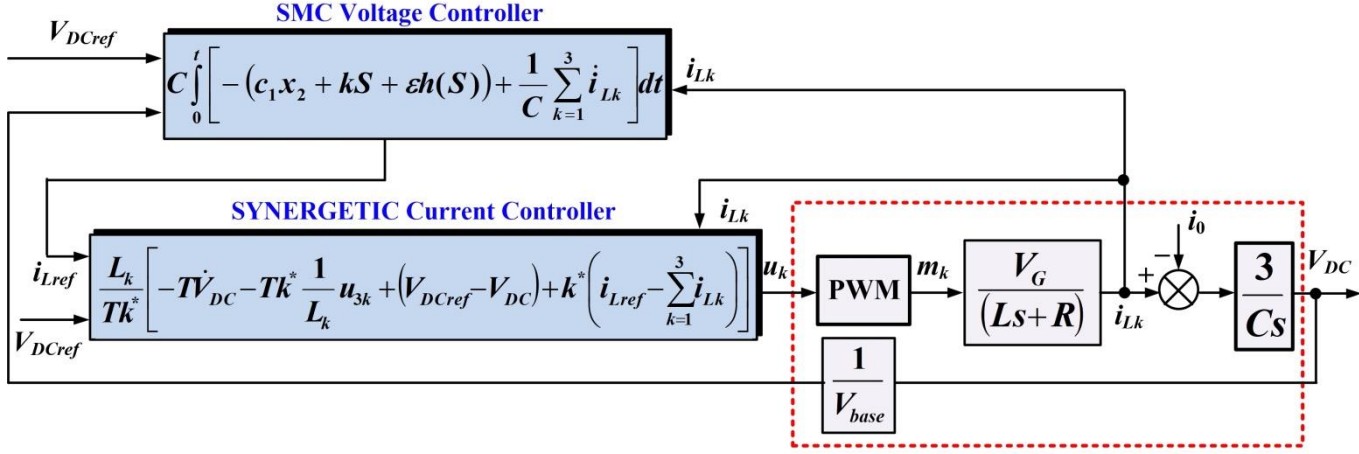

**Figure 10.** Block diagram modeled using the transfer function for the control system of the power electronic converter using an SMC controller for the voltage outer control loop and a synergetic controller for the current inner control loop.

For the synthesis of a fractional synergetic controller using the algorithm presented above, we choose a macro-variable Ψ with the following form:

$$\psi = D^\mu x_1 + k x_2 \tag{69}$$

By deriving the macro-variable Ψ, we obtain the following relation:

$$\dot{\psi} = D^\mu \dot{x}_1 + k^* \dot{x}_2 = -D^\mu \dot{V}_{DC} - k^* \sum_{k=1}^{3} \dot{i}_{Lk} \tag{70}$$

Accordingly, by adapting Equation (55) for the synthesis of the FO synergetic controller, we obtain the following relation:

$$T\left(-D^\mu \dot{V}_{DC} - k^* \dot{i}_L\right) + D^\mu \left(V_{DCref} - V_{DC}\right) + k^* \left(i_{Lref} - \sum_{k=1}^{3} i_{Lk}\right) = 0 \tag{71}$$

Using Equation (65), Equation (71) becomes:

$$-TD^{\mu+1}V_{DC} - Tk^* \frac{1}{L_k}(u_{3k} + u_k) + D^\mu \left(V_{DCref} - V_{DC}\right) + k^* \left(i_{Lref} - \sum_{k=1}^{3} i_{Lk}\right) = 0 \tag{72}$$

After rearranging the terms in Equation (72), we can write:

$$Tk^* \frac{1}{L_k} u_k = -TD^{\mu+1}V_{DC} - Tk^* \frac{1}{L_k} u_{3k} + D^\mu \left(V_{DCref} - V_{DC}\right) + k^* \left(i_{Lref} - \sum_{k=1}^{3} i_{Lk}\right) \tag{73}$$

The control law $u_k$ for k = 1, 2, 3, in the case of the FO synergetic controller, can be written as:

$$u_k = \frac{L_k}{Tk^*} \left[ -TD^{\mu+1}V_{DC} - Tk^* \frac{1}{L_k} u_{3k} + D^\mu \left(V_{DCref} - V_{DC}\right) + k^* \left(i_{Lref} - \sum_{k=1}^{3} i_{Lk}\right) \right] \tag{74}$$

Figure 11 presents the concise block diagram modeled using the transfer function for the control system of the power electronic converter using an FO SMC controller for the outer voltage control loop and an FO synergetic controller for the inner current control loop.

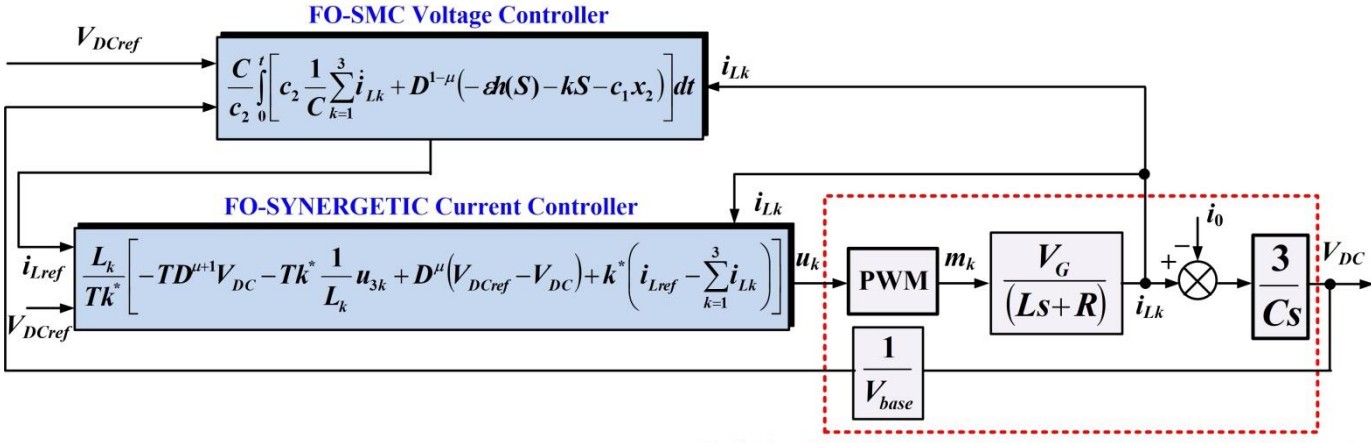

**Figure 11.** Block diagram modeled using the transfer function for the control system of the power electronic converter using an FO SMC controller for the voltage outer control loop and an FO synergetic controller for the current inner control loop.

To compare the performance of the control structures shown in Figures 8 and 9, they are named the SMC controller and synergetic controller and FO SMC controller and FO synergetic controller, respectively, for short.

*4.3. RL-TD3 Agent for Adjustment of the Command Signals of the FO SMC and FO Synergetic Controllers*

Among the types of machine learning, RL—in particular, the RL-TD3 agent—is the best suited for training for the control of an industrial process. Roughly speaking, the role of the RL-TD3 agent is to learn how to execute a task under the conditions of interacting with an unknown process, without requiring the explicit programming of the learning mode but only based on observations of the system and the performance of actions affecting the system in such a way as to maximize the cumulative reward. In the learning process of the RL-TD3 agent, an analogy with the control system of an industrial process can be made in the sense that observations are analogous to the reading of input quantities, actions are analogous to the provision of output quantities, and the cumulative reward can be associated with an integral optimization criterion. Among the components of the RL-TD3 agent, we can mention the policy component and the learning algorithm component. Figure 12 shows a block diagram of the components of an RL-TD3 agent.

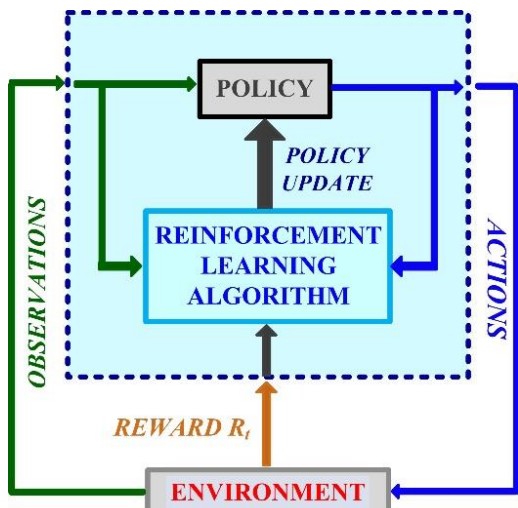

**Figure 12.** The bloc diagram of the components of the RL-TD3 agent.

The policy component can be described in terms of the way it associates actions with observations of the process and, in the case of the RL-TD3 agent, it is similar to the description of the functionality of a controller. The main goal of the learning algorithm is to find an optimal policy, which is achieved through iterative adaption of all the parameters of the policy component in such a way as to maximize the cumulative reward.

The main stages of the RL process include:

- Problem formulation—definition of the learning agent and how it interacts with the controlled process;
- Process creation— definition of the dynamic model and the interface associated with the controlled process;
- Reward creation— definition of the mathematical expression of the cumulative reward;
- Agent training—training of the agent according to the policy based on cumulative reward and definition of the learning algorithm;
- Agent validation—evaluation of the performance following the training stage;
- Policy implementation—implementation of the trained agent in a control system (e.g., by generating executable code for programming an embedded system).

Based on the operating mode of the RL-TD3 agent, we can specify that, after the training stage, it is capable of providing correction signals that overlap with the control signals provided by the FO SMC controller or FO synergetic controller, respectively, thus achieving superior performance in the control of the power electronic converter.

Figure 13 shows the proposed block diagram for the global control system of the power electronic converter using an FO SMC controller for the voltage outer control loop and an FO synergetic controller for the current inner control loop combined with the RL-TD3 agent.

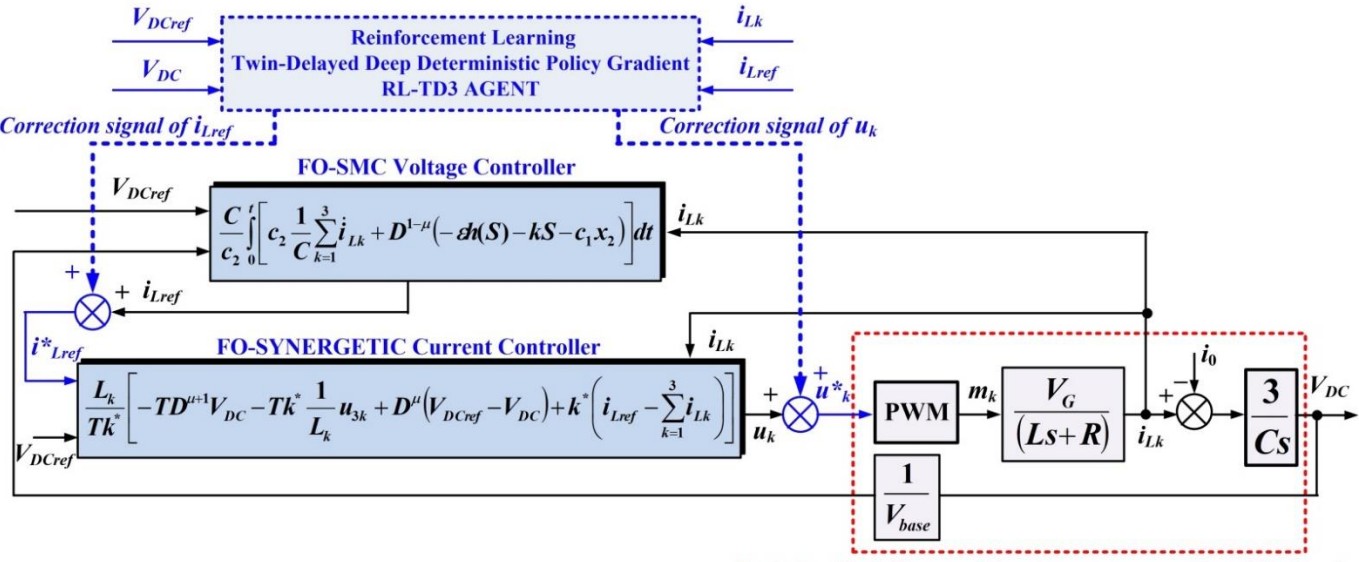

**Figure 13.** Block diagram modeled using the transfer function for the control system of the power electronic converter using an FO SMC controller for the voltage outer control loop and an FO synergetic controller for the current inner control loop combined with the RL-TD3 agent.

## 5. Numerical Simulations

In this section, based on the Matlab/Simulink R2021b development–simulation environment, we describe the comparison of the performance of the control systems of the power electronic converter considered as a benchmark, the parameters and the mathematical and transfer-function modeling of which are presented in Table 1 and in Section 2, respectively. The types of controllers and their parametrizations are presented in Sections 3 and 4, and details on the parameterizations are presented below. We should specify that the main task

of the power electronic converter control system is to maintain a constant DC voltage on DC bus 2 from the DC microgrid $V_{DCref}$ = 400 V.

The performances were compared in terms of the response time, steady-state error, overshoot, and error ripple in the DC voltage $V_{DCerror} = V_{DCref} - V_{DC}$. Table 2 summarizes the results obtained from the comparison of the control systems in relation to the benchmark presented. The calculation relation for the DC voltage $V_{DC}$ ripple is as follows [19]:

$$V_{DCripple} = \sqrt{\frac{1}{N}\sum_{i=1}^{N}\left(V_{DC}(i) - V_{DCref}(i)\right)^2} \tag{75}$$

where $N$ denotes the number of samples, $V_{DC}$ denotes the DC voltage for DC bus 2, and $V_{DCref}$ = 400 V.

**Table 2.** Performance indices for the power electronic converter control systems based on the proposed controllers.

| Controllers of the Power Electronic Converter | | Stationary Error (%) | Response Time (ms) | Overshoot (%) | $V_{DC}$ Ripple (V) |
|---|---|---|---|---|---|
| Controller for Voltage Outer Control Loop | Controller for Current Inner Control Loop | | | | |
| PI-Gao | PI-Gao | 0.1 | 11.8 | 0.1 | 48.79 |
| PI-Gamma | PI-Gao | 0.01 | 7.4 | 10 | 50.75 |
| FO PI | PI-Gao | 0.01 | 2.56 | 0.1 | 43.24 |
| FO TID | PI-Gao | 0.01 | 2.53 | 0.1 | 43.01 |
| FO lead-lag | PI-Gao | 0.01 | 2.48 | 0.1 | 42.92 |
| SMC | PI-Gao | 0.01 | 2.40 | 0.01 | 40.25 |
| FO SMC | PI-Gao | 0.01 | 2.27 | 0.01 | 40.07 |
| SMC | Synergetic | 0.01 | 2.08 | 0.01 | 39.94 |
| FO SMC | FO synergetic | 0.01 | 2.04 | 0.01 | 39.81 |
| FO SMC Combined with RL-TD3 agent | FO synergetic | 0.01 | 2.02 | 0.01 | 39.59 |

Figure 14 shows the schematic Matlab/Simulink R2021b implementation for the comparison of the performance of the control systems of the power electronic converters based on the PI-Gao, PI-gamma, FO PI, FO TID, and FO lead-lag voltage controllers. We specify that these controllers are for the outer voltage control loop, while for the inner current control loop, we use the parameterizations from Equations (13) and (14) for PI-type controllers, the inner current loop thus being approximated with a transfer function of the first order.

We can note that the PI-Gao, PI-gamma, and FO PI controllers have been previously used for comparisons with the same benchmark [18,19], and in this article the rest of the controllers presented are additionally synthesized for comparison with the same benchmark. The parameter values for the PI-Gao-, PI-Gamma-, and FO PI-type controllers are presented in detail in [18,19]. Matlab R2021b and the FOMCON toolbox were used to set the controller parameters for the FO TID and FO lead-lag controllers. The tuning of PI controllers using Ziegler–Nichols methods is a well-known technique. In the fractional case, the FOMCON toolbox for the Matlab R2021b utility program was used for the tuning of the FO PI controllers.

The results of the numerical simulations performed in Matlab/Simulink R2021b of the time evolution of the $V_{DC}$ voltage in the comparison of the PI-Gao, PI-gamma, FO PI, FO TID, and FO lead-lag voltage controllers for the control system of the power electronic converter are presented in Figure 15. Significant improvements can be noted for the FO PI, FO TID, and FO lead-lag controllers compared to the PI-Gao and PI-gamma controllers in terms of the performance of the power electronic converter control.

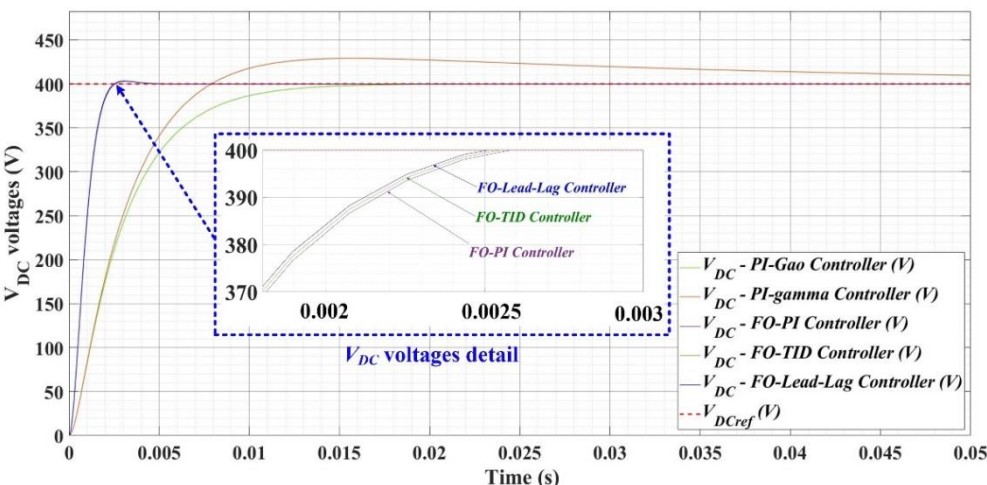

**Figure 14.** Matlab/Simulink implementation for the comparison of the control systems of power electronic converters using PI-Gao, PI-gamma, FO PI, FO TID, and FO lead-lag voltage controllers.

**Figure 15.** $V_{DC}$ voltage time evolution in the comparison of the PI-Gao, PI-gamma, FO PI, FO TID, and FO lead-lag voltage controllers for the control system of the power electronic converter.

Next, we present the comparative results for the control systems of the power electronic converters based on the controllers synthesized in Section 4 (i.e., the SMC and synergetic controllers of integer or fractional orders) and indicate the possibility of improving the control system by using an RL-TD3 agent. Figure 16 shows the schematic Matlab/Simulink R2021b implementation for the most complex control structures proposed in this article; i.e., the FO SMC controller for the voltage outer control loop and the FO synergetic controller for the current inner control loop combined with a RL-TD3 agent.

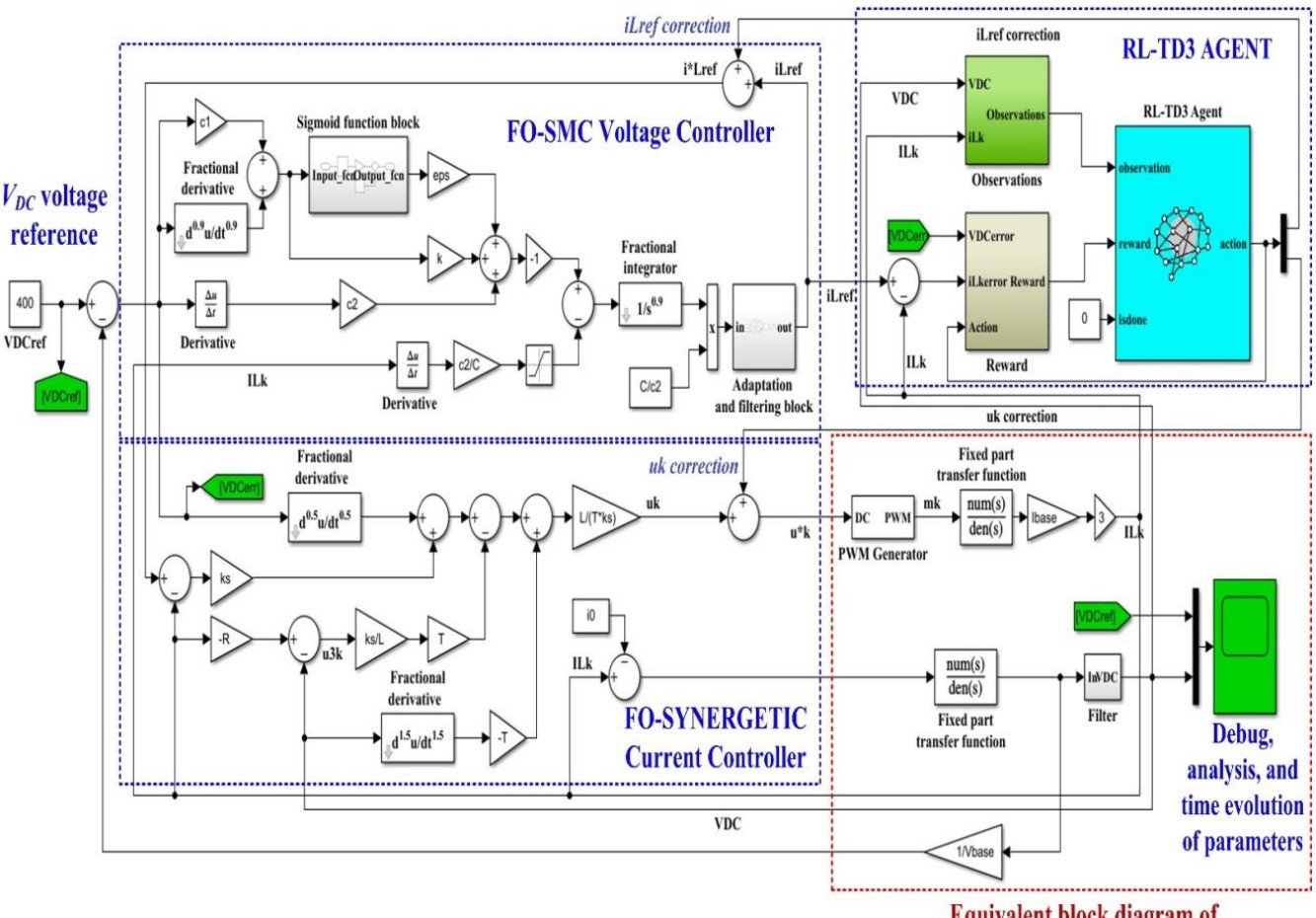

**Figure 16.** Matlab/Simulink R2021b implementation for the control systems of the power electronic converter using the FO SMC controller for the voltage outer control loop and the FO synergetic controller for the current inner control loop combined with the RL-TD3 agent.

For the RL-TD3 agent described in Section 4, the cumulative reward used is given in Equation (76), and Figure 17 summarizes the results for the RL-TD3 agent training stage. Once the RL-TD3 agent is trained, it can provide correction signals for the control signals of the FO SMC voltage controller and FO synergetic current controller in order to achieve superior control performance with the presented benchmark.

The FO SMC controller parameters used in the numerical simulations in Matlab/Simulink R2021b, according to the notations in Section 4, are as follows: $c_1 = 0.1$, $c_2 = 0.1$, $k = 118{,}000$, and $\varepsilon = 110$.

The FO synergetic controller parameters used in the numerical simulations in Matlab/Simulink, according to the notations in Section 4, are as follows: $k^* = 10{,}000$, $T = 3$, and $\mu = 0.55$.

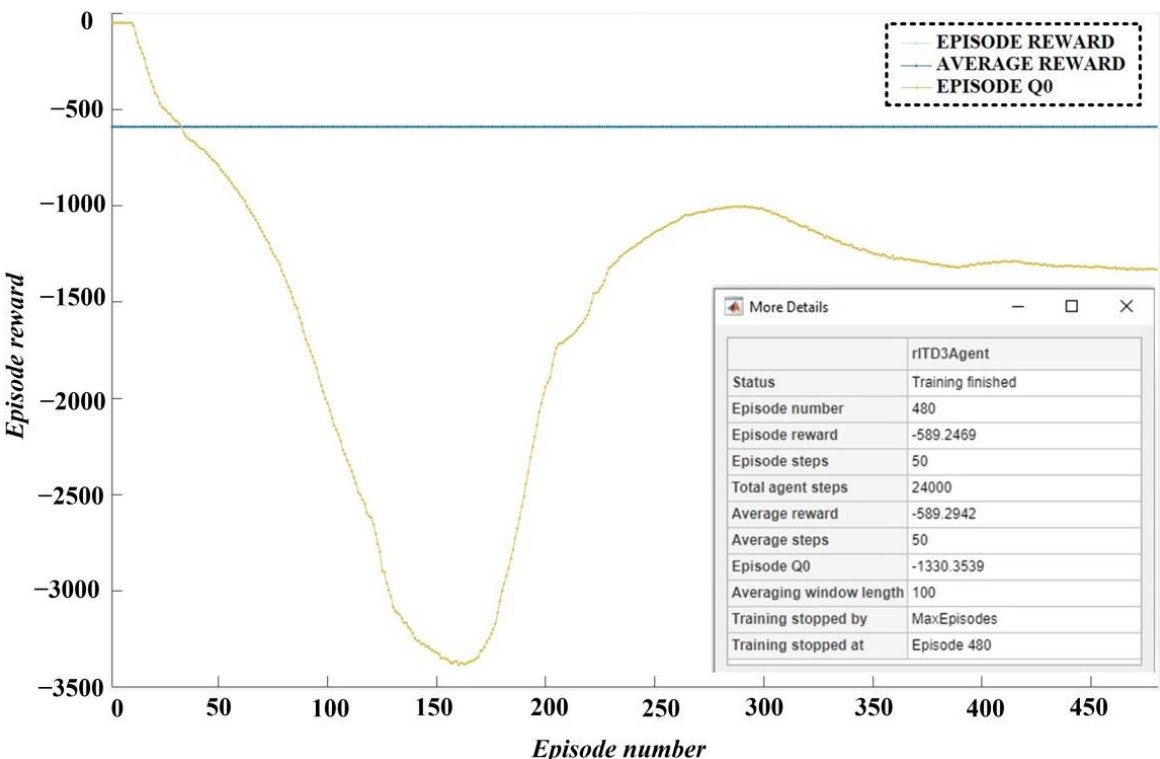

**Figure 17.** Reward evolution during the training stage of the RL-TD3 agent for the command-signal-correction control systems of the power electronic converter using the FO SMC controller for the voltage outer control loop and the FO synergetic controller for the current inner control loop.

The optimization criterion (the reward) used in the training stage for the control system for the power electronic converter based on the FO SMC and FO synergetic controllers combined with the RL-TD3 agent is the following:

$$r_{FO-SMC\&FO-SYNERGETIC} = -\left(5i^2_{Lerror} + 5V^2_{DCerror} + 0.1\sum_j \left(u^j_{t-1}\right)^2\right) \tag{76}$$

where $u^j_{t-1}$ represents the actions from the previous step.

For the parameterizations shown in Figure 18, in order to note the positive evolution in terms of the performance of the controllers proposed in Section 4, we present a comparison of the results of the numerical simulations performed in Matlab/Simulink R2021b of the time evolution of the $V_{DC}$ voltage for the benchmark presented using the following combinations of proposed controllers: FO PI voltage controller and PI-Gao current controller, SMC voltage controller and PI-Gao current controller, FO SMC voltage controller and PI-Gao current controller, SMC voltage controller and synergetic current controller, FO SMC voltage controller and FO synergetic current controller, FO SMC voltage controller and FO synergetic current controller combined with the RL-TD3 agent, and FO PI voltage controller and PI-Gao current controller.

As shown in Figure 19, it can be noted that, if the current absorbed by the consumers connected to DC bus 2 in the DC microgrid has a step change of 100 A, the control system of the power electronic converter based on the FO SMC and FO synergetic controllers combined with the RL-TD3 agent has a very good response, and the only difference, compared to the case where $i_0 = 0$ A, is a slight increase in the overshoot from 0.01% to 0.12%. This proves that the proposed control system, in addition to the top performances summarized in Table 2, ensures good parametric robustness.

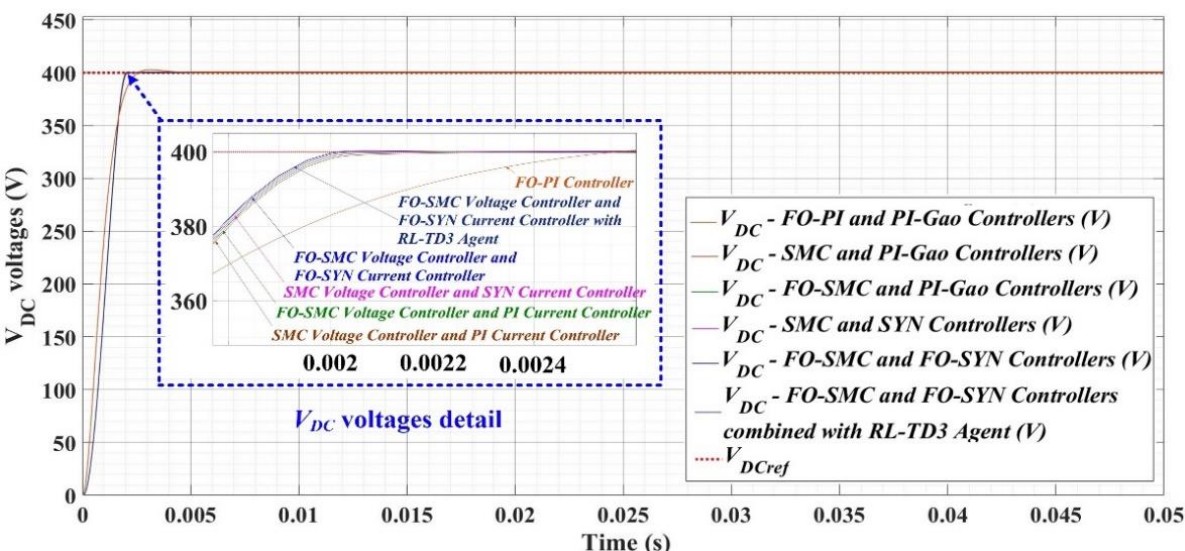

**Figure 18.** $V_{DC}$ voltage time evolution from the comparison of the FO PI and PI-Gao controllers, SMC and PI-Gao controllers, FO SMC and PI-Gao controllers, SMC and synergetic controllers, FO SMC and FO synergetic controllers, and FO SMC and FO synergetic controllers combined with the RL-TD3 agent for the control system of the power electronic converter.

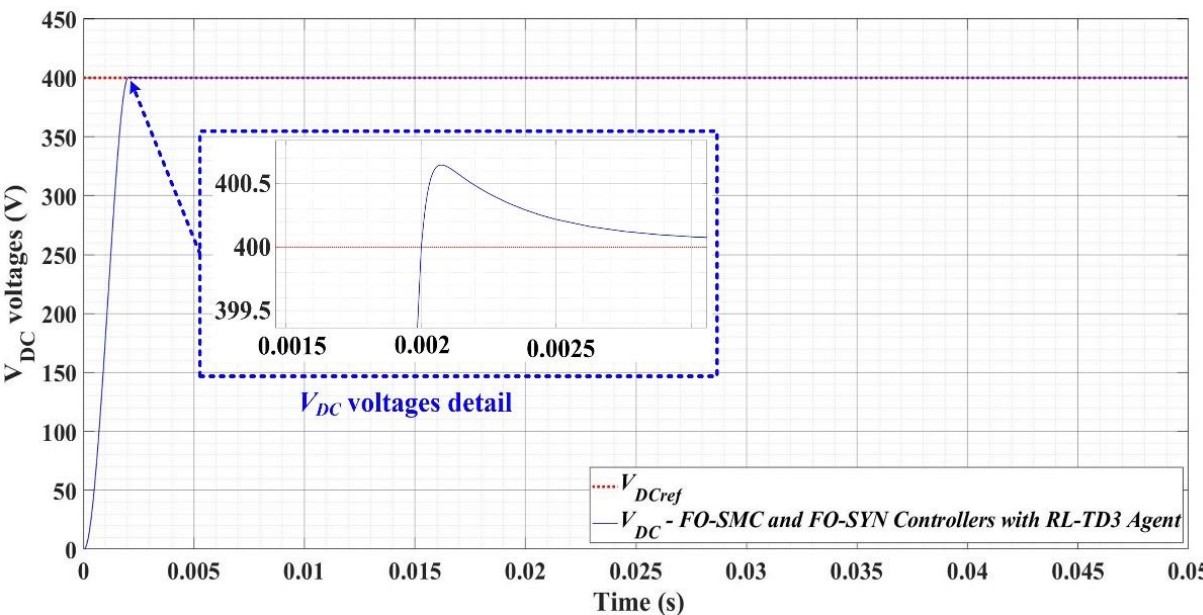

**Figure 19.** $V_{DC}$ voltage time evolution based on FO SMC and FO synergetic controllers combined with RL-TD3 agent for control system of the power electronic converter in the case of a DC microgrid current for DC bus 2 $i_0 = 100$ A.

The current $i_0$, which represents the current absorbed by the consumers, represents the main disturbance in the system. Thus, in Figure 19, we can see a very good response from the control system in the case of a jump of 100 A, which proves the good robustness of the proposed controller.

This brief analysis of the performance of the control systems of the power electronic converter presented as a benchmark shows that the performance of the controllers proposed in this paper significantly improves the performance described in [18,19].

Thus, a first leap in terms of improving the performance of the control system compared to the use of classical PI controllers is provided by the use of FO PI, FO TID, and FO

lead-lag fractional controllers. A second leap in terms of improving the performance of the control system—in particular, the response time and $V_{DC}$ ripple—compared to the use of the FO PI, FO TID, and FO lead-lag fractional controllers is provided by the use of the controllers synthesized in Section 4, among which the FO SMC voltage controller and FO synergetic current controller combined with the RL-TD3 agent have the best performance.

## 6. Conclusions

This article presented the topology and mathematical modeling using differential equations and transfer functions of a three-phase power electronic converter providing the interface and interconnection between the grid and a DC microgrid. This system was used as a benchmark in a series of studies on the performance of the control systems used, the main task of which was to maintain the DC voltage supplied to the microgrid at an imposed constant value, regardless of the total value of the current absorbed by the consumers connected to the DC microgrid. We presented fractional calculus elements that were used to synthesize a first set of FO PI, FO TID, and FO lead-lag controllers that significantly improved the performance of the control system of the power electronic converter compared to classical PI controllers. The next set of proposed and synthesized controllers were based on SMC, together with its more general and flexible synergetic control variant, and both integer-order and FO controllers were used. The proposed control structures were cascade control structures, combining the SMC properties of robustness and control over nonlinear systems for the outer voltage control loop with properly tuned synergetic controllers enabling faster response time for the inner current control loop. To achieve superior performance, this type of cascade control also used a properly trained RL-TD3 agent, which provided correction signals overlapping with the command signals of the current and voltage controllers. We presented the Matlab/Simulink R2021b implementations of the synthesized controllers and RL-TD3 agent, along with the results of numerical simulations performed for the comparison of the performance of the control structures. It can be concluded that the structure here proposed by the authors (FO SMC controller for outer control loop and FO synergetic controller for inner control loop combined with an RL-TD3 agent)—which has also been described in other articles but with other benchmarks [24,25]—provided top performance in the case of the benchmark used in this article, and it can be recommended for cases where the control can be cascaded on two levels. In [25], due to the particularity of the benchmark, the authors also carried out real-time implementation of the proposed control system in an embedded system. In future approaches, by using an RT-Opal system, we also intend to carry out real-time implementation with the benchmark used in this article.

**Author Contributions:** Conceptualization, M.N.; methodology, M.N.; software, M.N. and C.-I.N.; validation, M.N. and C.-I.N.; formal analysis, M.N. and C.-I.N.; investigation, M.N.; resources, M.N.; data curation, M.N. and C.-I.N.; writing—original draft preparation, M.N. and C.-I.N.; writing—review and editing, M.N. and C.-I.N.; visualization, M.N. and C.-I.N.; supervision, M.N.; project administration, M.N.; funding acquisition, M.N. All authors have read and agreed to the published version of the manuscript.

**Funding:** This work was developed with funds from the Ministry of Research, Innovation and Digitization of Romania as part of the NUCLEU Program: PN 19 38 01 03.

**Institutional Review Board Statement:** Not applicable.

**Informed Consent Statement:** Not applicable.

**Data Availability Statement:** Not applicable.

**Conflicts of Interest:** The authors declare no conflict of interest.

## Appendix A

To obtain the integer transfer functions for the FO PI, FO TID, and FO lead-lag controllers in continuous and discrete form, respectively, we first used the FOMCON toolbox with $\omega = [10^{-2}; 10^3]$ rad/s and then Tustin substitution. The transfer functions s and, respectively, the functions in the z-domain are presented in Figures A1–A3.

Based on the form of the transfer function in the $z$-domain, we can easily obtain a finite-difference time-domain equation suitable for real-time implementation in an embedded system.

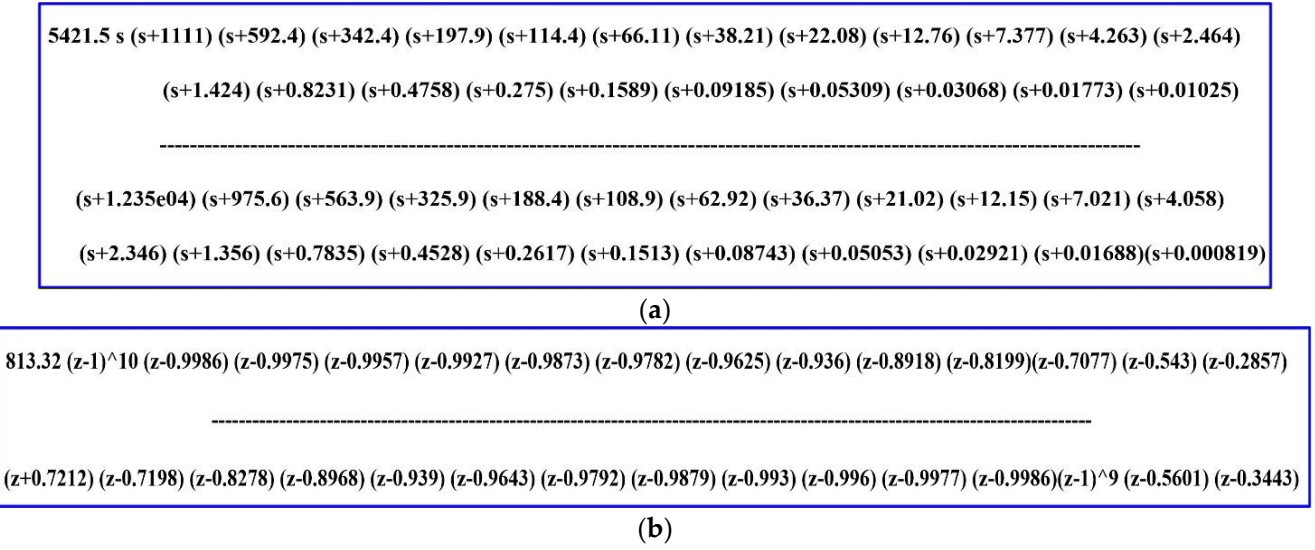

**(a)**

**(b)**

**Figure A1.** Transfer functions for the FO PI controller: (**a**) continuous form; (**b**) discrete form.

6517.8 (s+1114) (s+593.3) (s+343.3) (s+198.8) (s+115.3) (s+67.1) (s+39.36) (s+23.68) (s+15.48) (s+10.7) (s+6.693) (s+3.969)

(s+2.318) (s+1.347) (s+0.7806) (s+0.4519) (s+0.2614) (s+0.1512) (s+0.08739) (s+0.05052) (s+0.0292) (s+0.01688) (s+0.000818)

-------------------------------------------------------------------------------------------------------------------------------

s (s+1.235e04) (s+975.6) (s+563.9) (s+325.9) (s+188.4) (s+108.9) (s+62.92) (s+36.37) (s+21.02) (s+12.15) (s+7.021) (s+4.058)

(s+2.346) (s+1.356) (s+0.7835) (s+0.4528) (s+0.2617) (s+0.1513) (s+0.08743) (s+0.05053) (s+0.02921) (s+0.01688) (s+0.000819)

**(a)**

0.49399 (z-0.2845) (z-0.5424) (z-0.707) (z-0.8192) (z-0.891) (z-0.9351) (z-0.9614) (z-0.9766) (z-0.9846) (z-0.9894) (z-0.9933) (z-0.996) (z-0.9977) (z-1)^10 (z+1)

-------------------------------------------------------------------------------------------------------------------------------

(z+0.7212) (z-0.7198) (z-0.8278) (z-0.8968) (z-0.939) (z-0.9643) (z-0.9792) (z-0.9879) (z-0.993) (z-0.996) (z-0.9977) (z-0.9986) (z-1)^10 (z-0.5601) (z-0.3443)

**(b)**

**Figure A2.** Transfer functions for the FO TID controller: (**a**) continuous form; (**b**) discrete form.

$$
\begin{aligned}
&1.8023\ (s+1235)\ (s+1235)\ (s+1111)\ (s+1389)\ (s+803.1)\ (s+781.4)^2\ (s+719.7)\ (s+464.2)\ (s+451.6)\ (s+451.6)\ (s+415.9)\ (s+268.3) \\
&(s+261)\ (s+261)\ (s+240.4)\ (s+155.1)\ (s+150.9)\ (s+150.9)\ (s+138.9)\ (s+89.62)\ (s+87.19)\ (s+87.18)\ (s+80.29)\ (s+51.79) \\
&(s+50.39)\ (s+50.38)\ (s+46.4)\ (s+29.94)\ (s+29.13)\ (s+29.12)\ (s+26.81)\ (s+17.3)\ (s+16.83)\ (s+16.82)\ (s+15.5)\ (s+10) \\
&(s+9.73)\ (s+9.717)\ (s+8.969)\ (s+5.78)\ (s+5.623)\ (s+5.61)\ (s+5.214)\ (s+3.34)\ (s+3.25)\ (s+3.233)\ (s+3.075)\ (s+1.931) \\
&(s+1.878)\ (s+1.116)\ (s+1.086)\ (s+0.6449)\ (s+0.6275)\ (s+0.3728)\ (s+0.3714)\ (s+0.3642)\ (s+0.3627)\ (s+0.2154)\ (s+0.2152) \\
&(s+0.21)\ (s+0.2096)\ (s+0.1245)\ (s+0.1245)\ (s+0.1213)\ (s+0.1212)\ (s+0.07197)\ (s+0.07196)\ (s+0.07006)\ (s+0.07002) \\
&(s+0.0416)\ (s+0.04159)\ (s+0.04048)\ (s+0.04047)\ (s+0.02404)^2\ (s+0.02339)\ (s+0.02339)\ (s+0.01389)^2\ (s+0.01352)\ (s+0.01352) \\
&(s+0.00018)^2\ (s+9e\text{-}05)^2\ (s^2+1.27s+0.4033)\ (s^2+2.18s+1.189)\ (s^2+3.709s+3.44)\ (s^2+0.151s+3.511) \\
&\rule{14cm}{0.4pt} \\
&(s+1389)\ (s+1235)\ (s+1234)\ (s+1111)\ (s+803.1)\ (s+781.4)\ (s+781.3)\ (s+719.6)\ (s+464.2)\ (s+451.6)\ (s+451.6)\ (s+415.9)\ (s+268.3) \\
&(s+261)\ (s+261)\ (s+240.4)\ (s+155.1)\ (s+150.9)\ (s+150.8)\ (s+138.9)\ (s+89.62)\ (s+87.19)\ (s+87.17)\ (s+80.26)\ (s+51.79) \\
&(s+50.39)\ (s+50.37)\ (s+46.36)\ (s+29.94)\ (s+29.13)\ (s+29.1)\ (s+26.77)\ (s+17.3)\ (s+16.83)\ (s+16.8)\ (s+15.44)\ (s+10) \\
&(s+9.73)\ (s+9.693)\ (s+8.89)\ (s+5.78)\ (s+5.623)\ (s+5.581)\ (s+5.094)\ (s+3.34)\ (s+3.25)\ (s+3.196)\ (s+2.87)\ (s+1.931) \\
&(s+1.878)\ (s+1.793)\ (s+1.281)\ (s+1.116)\ (s+1.086)\ (s+0.9373)\ (s+0.6685)\ (s+0.6449)\ (s+0.6275)\ (s+0.3728)\ (s+0.3627) \\
&(s+0.2154)\ (s+0.2096)\ (s+0.1245)\ (s+0.1241)\ (s+0.1231)\ (s+0.1212)\ (s+0.07197)\ (s+0.07194)\ (s+0.07052)\ (s+0.07002) \\
&(s+0.0416)\ (s+0.04159)\ (s+0.04062)\ (s+0.04047)\ (s+0.02404)\ (s+0.02404)\ (s+0.02343)\ (s+0.02339)\ (s+0.01389)^2\ (s+0.01353) \\
&(s+0.01352)\ (s+0.00018)^2\ (s+9.001e\text{-}05)\ (s+9e\text{-}05)\ (s^2+0.4309s+0.04642)\ (s^2+0.7617s+0.1451)\ (s^2+1.274s+0.4438)\ (s^2+2.948s+2.348)
\end{aligned}
$$

(**a**)

$$
\begin{aligned}
&(z-0.2367)\ (z-0.2366)\ (z-0.2857)\ (z-0.1803)\ (z-0.427)\ (z-0.4381)^2\ (z-0.4708)\ (z-0.6233)\ (z-0.6316)^2\ (z-0.6557)\ (z-0.7635) \\
&(z-0.7691)^2\ (z-0.7854)\ (z-0.8561)\ (z-0.8597)^2\ (z-0.8701)\ (z-0.9142)\ (z-0.9165)^2\ (z-0.9228)\ (z-0.9495)\ (z-0.9508)^2 \\
&(z-0.9547)\ (z-0.971)^3\ (z-0.9735)\ (z-0.9835)^4\ (z-0.9904)^4\ (z-0.9945)^4\ (z-0.9968)^4\ (z-0.9981)^4\ (z-0.9989)^4 \\
&(z-0.9994)^4\ (z-1)^{32}\ (z^2-2z+0.9998) \\
&\rule{14cm}{0.4pt} \\
&(z-0.1803)\ (z-0.2366)\ (z-0.2367)\ (z-0.2857)\ (z-0.427)\ (z-0.4381)\ (z-0.4382)\ (z-0.4708)\ (z-0.6233)\ (z-0.6316)\ (z-0.6316) \\
&(z-0.6557)\ (z-0.7635)\ (z-0.7691)\ (z-0.7691)\ (z-0.7854)\ (z-0.8561)\ (z-0.8597)\ (z-0.8597)\ (z-0.8701)\ (z-0.9142)\ (z-0.9165) \\
&(z-0.9165)\ (z-0.9228)\ (z-0.9495)\ (z-0.9508)\ (z-0.9509)\ (z-0.9547)\ (z-0.971)^3\ (z-0.9736)\ (z-0.9835)^4\ (z-0.9905)^4 \\
&(z-0.9945)^4\ (z-0.9968)^4\ (z-0.9981)^3\ (z-0.9989)^4\ (z-0.9994)^5\ (z-1)^{32}\ (z^2-1.997z+0.9971)
\end{aligned}
$$

(**b**)

**Figure A3.** Transfer functions for the FO lead-lag controller: (**a**) continuous form; (**b**) discrete form.

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
