# Peer review of "Improved Performance in the Control of DC-DC Three-Phase Power Electronic Converter Using Fractional-Order SMC and Synergetic Controllers and RL-TD3 Agent"

_fractalfract, doi:10.3390/fractalfract6120729_

Round 1
Reviewer 1 Report
The paper under review addresses the topology, mathematical modeling and transfer functions of a three-phase power electronic converter which provides the interface and interconnection between the grid and a DC microgrid. Authors have adopted fractional elements that are used to synthesize a set of controllers. This thematic is relevant and interesting. Nonetheless, there are several weakness in this paper including:
1) Abstract should be improved. One should clearly point out the motivation, objective, adopted/developed methods, and finally the mains achieved results. There are several acronyms and some of them (e.g., 'PI' is not identified). Although PI is an acronym known by experts/researchers in the field of control systems, it is necessary to explain/declare all acronyms presented in a scientific text. Overuse of acronyms makes reading unpleasant and harder to understand. As is, the Abstract is very confusing. I suggest that it be rewritten.
2) For a more correlation of the present study to the problem statement/motivation, the 'Introduction' section can be better described, expanded and more detail should provided about the gap in the literature. In other words, the main goal and the motivation of this paper are not well justified. The novelty and contribution of the manuscript are also not clear enough. Moreover, the literature review is not adequate. I suggest the authors keep up to date the references by citing seminal related works. Multiple papers related to the fractional order controllers applications have been published and there should be cited.
3)
a)The proposed mathematical model is based on simple differential Equation and it lacks novelty. I can't realize a high level of mathematical and/or computational method being employed.
b) The software (MATLAB/SIMULINK) used is declared. However, the choice of the values of the parameters also deserves more comments and should be clearly justified and discussed.
I suggest that material and methods be also described in sufficient detail so that another researcher is able to reproduce the findings described.
4) I also suggest that some Figures such as the Figs. 6, 7 and 8 (Transfer Functions equations) be moved to an Appendix.
5) I recommend add more criteria to improve the arguments about the fit results.
6) Moreover, a robust performance evaluation of the proposed model is also missing. In other words, comparison (including statistical treatment) between the proposed method and existing approaches published in the literature is imperative in order to certify the efficiency of the method proposed by authors in this study.
7) Thereafter, one should provide a deeper discussion about the achieved results.
8) Additional proofreading is needed to fix some typos.
The paper needs major revisions before being accepted.
Author Response
Dear reviewer, thanks for your recommendations.
1), 2), and 3) In the Abstract it is specified that it starts from a benchmark. Details about the benchmark are given in the Introduction and in the other sections. The acronym PI was specified. We consider that thus, the Abstract is well structured. (By the way, this is also the opinion of the other two reviewers). In this article, starting from a benchmark used in references [18,19], which consists of a DC-DC three-phase power electronic converter, and which allows comparing the performance of the control systems of this type of converter, when the main task of the control systems is to maintain the constant voltage supplied to the DC microgrid under the conditions of a variable consumption required by it, we propose and synthesize a series of controllers with which superior performance is obtained for the benchmark presented. In order to improve the performance of the control systems used on the mentioned benchmark, in this article, starting from the definitions and particular structures of the fractional calculus, we propose and synthesize a series of fractional and integer controllers, as well as combined controllers to be used in voltage outer loop control and current inner loop control [24,25]. A properly trained RL-TD3 agent combined with a cascade control structure is also used to improve this performance [26,27]. The main performance indicators covered by the comparison of power electronic converter control systems are: steady-state error, overshoot, response time and ripple of the DC voltage supplied to the DC microgrid. The specified benchmark was published in ISI Web of Science papers (Journal and Conferences). Thus, the low importance of the subject is a subjective opinion.
Among the main contributions brought in this article we can mention:
- Mathematical modeling by differential equations and transfer functions of the power electronic converter which provides the interface and interconnection between the grid and a DC microgrid, a system that is used as a benchmark;
- Synthesis of fractional controllers using Matlab and FOMCON toolbox [28-30], i.e.: FO-PI, FO-TID, and FO-Lead-Lag for the presented benchmark;
- Synthesis of both integer and fractional SMC and Synergetic controllers for the presented benchmark;
- Implementation in Matlab/Simulink of a control structure proposed by the authors consisting of FO-SMC controllers for voltage outer loop control and FO-Synergetic for current inner loop control operating in tandem with an RL-TD3 agent to achieve superior control performance of the presented benchmark;
- Implementation in Matlab/Simulink of the synthesized controllers in order to compare the performance of the control system of the presented benchmark.
The choice of parameter values for the PI-Gao, PI-Gamma, and FO-PI type controllers are presented in detail in the articles presented in [18,19]. Matlab and FOMCON toolbox are used to set the controller parameters: FO-TID and FO-Lead-Lag. The tuning of PI controllers by using Ziegler–Nichols methods is a well-known technique. In the fractional case, the FOMCON toolbox for the MATLAB utility program is used for the tuning of FO-PI controllers. In order to obtain the optimal tuning parameters in the fractional case, a number of optimization methods are incorporated in the FOMCON toolbox, both in the frequency range and in the time domain. In the frequency range, the goal of optimizing the parameters is achieved by obtaining optimal performance in terms of the sensitivity function S(jω) for disturbance rejection for the low and middle frequency range and the rejection of the high frequency noise using the complementary sensitivity function T(jω). In the time domain, the tuning of fractional controllers is carried out by minimizing optimal criteria, such as the integral absolute error. For SMC, FO-SMC, Synergetic, and FO-Synergetic controllers, in this article, calculations are presented in detail.
4) We have moved figures 6, 7, and 8 to the Appendix.
5), 6), and 7) A first leap in terms of improving the performance of the control system, compared to the use of classical PI controllers, is given by the use of FO-PI, FO-TID, and FO-Lead-Lag fractional controllers. A second leap in terms of improving the performance of the control system, in particular the response time and VDC ripple, compared to the use of the FO-PI, FO-TID, and FO-Lead-Lag fractional controllers is given by the use of the proposed controllers synthesized in Section 4, of which the FO-SMC voltage controller and FO-Synergetic current controller combined with the RL-TD3 agent have the best performance. In Table 2 are presented the performance indices of the power electronic converter control system based on the proposed controllers. In Figure 19, if the current absorbed by the consumers connected to the DC Bus 2 in the DC microgrid has a step change of 100A, it can be noted that the control system of the power electronic converter based on FO-SMC and FO-Synergetic controllers combined with RL-TD3 agent, has a very good response, and the only difference, compared to the case i0 = 0A, is a slight increase of the overshoot from 0.01% to 0.12%. This proves that the proposed control system, in addition to the top performances summarized in Table 2, ensures a good parametric robustness. The current i0, which represents the current absorbed by the consumers, represents the main disturbance on the system. Thus, in Figure 19 we can see a very good response of the control system in case of a jump of 100A, which proves a good robustness of the proposed controller.
8) For typos was made corrections.

Reviewer 2 Report
In this article, starting from a benchmark represented by a DC-DC three-phase power electronic converter used as an interface and interconnection between the grid and a DC microgrid. Authors have presented the topology, the mathematical modeling by differential equations and transfer functions of the DC-DC three-phase power electronic converter which provides the interface between the grid and a DC microgrid. The main task of the presented control systems is to maintain the DC voltage supplied to the microgrid to an imposed constant value, regardless of the total value of the current absorbed by the consumers connected to the DC microgrid. This manuscript is well written/organized and covers timely subject in the field. Hence, this manuscript could be accepted in the present form.
Author Response
Dear reviewer, thanks for your appreciations.
Reviewer 3 Report
The paper by Nicola & Nicola deals with the performance improvement of a power converter, in the framework of electronics. By means of mathematical modeling and numerical simulations, the roles of different implementations - with the aim of supplying a constant DC voltage to the microgrid independently of the load applied - are investigated. After recalling the Kirchhoff laws and Laplace transform, and expressing the transfer function in terms of the circuit parameters, the authors introduce few basic notions of fractional calculus (derivatives and integrals of non-integer order), and present its application to the problem under analysis. Focusing then on concepts such as machine\reinforcement learning, agent training\validation and synergetic controllers, they study the efficiency of several systems with respect to a classical benchmark, and conclude by identifying the best implementation method via computations with Matlab\Simulink. Their results are carefully presented and plotted, and the relevance of these findings with respect to the existing bibliography from scientific literature is thoroughly discussed.
In my opinion, the article deserves publication on Fractal and Fractional. Hereafter I only mention few minor issues, to be addressed by the authors.
1) Line 13: just as done in the following lines in the abstract, it would be useful to write in full the meaning of the PI acronym.
2) Line 126: "ÅŸi" -> "and"
3) Lines 311 and 328 and 386 and 402: is the expression "presents synthetic" correct?
4) Line 433: what does "LR" stand for?
Author Response
Dear reviewer, thanks for your recommendations and appreciations.
1) We have written the full meaning of the PI acronym in Abstract.
2) We have corrected in Line 126: "ÅŸi" -> "and".
3) We have inserted in Lines 311 and 328 and 386 and 402: "presents synthetic" with "presents concise".
4) We have corrected in Line 433: "LR" with "RL" (Reinforcement Learning).

Round 2
Reviewer 1 Report
Authors managed to address tho most of my questions.